# New Modification of Polar Nonlinear Optical Iodate Fluoride PbF(IO₃), the Family MX(IO₃), M = Bi, Ba, Pb, X = O, F, (OH) Related to Aurivillius Phases and Similar Iodates

Elena Belokoneva [1,*], Olga Reutova [1], Anatoly Volkov [1], Olga Dimitrova [1] and Sergey Stefanovich [2]

1   Department of Crystallography and Crystal Chemistry, Geological Faculty,
    Lomonosov Moscow State University, 119991 Moscow, Russia
2   Laboratory of Functional Materials, Chemical Faculty, Lomonosov Moscow State University,
    119991 Moscow, Russia
*   Correspondence: elbel@geol.msu.ru; Tel.: +7-495-939-4926

**Abstract:** A new modification of PbF(IO₃) has been obtained as single crystals from hydrothermal synthesis, alongside the known centrosymmetric, Pb(IO₃)₂, as a second phase. Measured with the Kurtz-Perry SHG method, the new crystals are phase-matchable for YAG:Nd laser radiation and demonstrate strong SHG output. According to an X-ray diffraction analysis conducted on a single crystal at low temperature, the new crystals appear monoclinic, of space group *Pn*, as opposed to the known orthorhombic modification of the PbF(IO₃), of space group *Iba*2. The new crystals were also measured at room temperature, showing an orthorhombic disordered variant of the new phase (space group *C2ma*, standard *Abm*2). This variant presents an "average structure" with the mirror plane in the group. The low-temperature X-ray single-crystal experiment allowed us to find the correct structural model, where the mirror plane was found as a twin element in the real monoclinic *Pn* structure. A careful crystal chemical analysis led to a whole family of nonlinear optical crystals with a common formula, AX(IO₃), A = Bi, Ba, Pb, X = O, F, (OH), currently counting six representatives, including the well-known BiO(IO₃). All of them possess common central cationic layers similar to those known in Aurivillius-type phases, with anionic iodate layers attached above and below these layers instead of the perovskite-like, or halogens. The structure–property relationships are discussed with respect to the important role of the large cations: Pb²⁺, Bi³⁺ or Ba²⁺. A number of iodates with similar structures are also analyzed.

**Keywords:** iodates; hydrothermal synthesis; second-order optical nonlinearity; single-crystal X-ray diffraction; Aurivillius phases; disorder; structure–property relations



## 1. Introduction

Multiple metal iodates have been obtained and investigated in recent years in the search for new materials containing polar groups (IO₃)⁻ favorable for optical nonlinearities. An I⁵⁺ lone pair of electrons completes the pyramidal umbrella-like configuration of the surrounding oxygens up to the tetrahedron and can be considered in line with the polar structural fragments in Pb²⁺ or Bi³⁺ oxides, also possessing lone pairs of electrons. These polar (IO₃)⁻ groups are the structural features that result in technically important properties, such as piezo- and pyro-electricity, and second-order optical nonlinearity, summarized in reviews [1,2].

Successful research for new phases in different groups of substances requires deeper understanding of crystal growth conditions combined with the fundamental structural data based on the topology and symmetry peculiarities responsible for special properties. In particular, single crystals often demonstrate a polytypic nature with order (O) and disorder (D) features, as well as twinning. A recent attempt to unravel the role of OD in the

structure–property relationship is made in [3] for the nonlinear optical iodate family, with the new $Rb_3Sc(IO_3)_6$ and other known members.

The Aurivillius phases are a family of oxides and oxyhalogenides which have attracted interest due to numerous ferroelectric and superconducting materials and compounds with magnetic, photocatalytic, and other valuable properties. The predecessor of these materials is $Bi_2NbO_5F$ [4]; however, it is useful to continue the comparison of structures and properties of this classical oxyhalogenide with its up-to-date analogs and related iodates.

In this review, the synthesis and structure of the new $PbF(IO_3)$ modification are investigated and used as a base for the separation of a number of iodate structures related to the Aurivillius phases. The analysis of the role of large cations in the properties is carried out. A number of iodates related by formula or structure are also analyzed.

## 2. New Modification of $PbF(IO_3)$

### 2.1. Materials and Methods

The new single crystals were synthesized under hydrothermal conditions from a mixture of chemically pure components of $PbF_2$ (1 g, 4 mmol), $Pb(NO_3)_2$ (3 g, 9.1 mmol), and $I_2O_5$ (5 g, 15 mmol). All the reagents were added to a Teflon-lined, stainless steel pressure vessel of 30 mL capacity, which was filled with distilled water up to 50%. The synthesis was carried out at 543 K (270 °C). The reaction took place to completion under heating for 5 days, followed by cooling to room temperature for over 24 h. The reaction products were washed with hot distilled water and dried at room temperature.

A morphological separation of the new crystals was done using an optical microscope with x32 magnification. Two types of crystals were found in the crystallization products: (I) transparent colorless, often fissured, prismatic, and (II) prismatic isometric, more perfect crystals of smaller size. The ratio of both forms was ~1:4. The prismatic crystals (I) were elongated up to 0.5 mm, with an average size of ~0.2 mm. They were among the mass of crystals (II) of smaller sizes, ~0.05–0.15 mm. The common yield of the experiment was close to ~90%.

Elemental analysis was carried out on the surfaces of a crystal using a Jeol JSM-6480LV scanning electronic microscope combined with a WDX analysis. The test revealed the presence of Pb and I for both types (I) and (II) of crystals.

The separated crystals (I) had an XRD pattern with no analogs in the ICDD database (Figure S1a in Supplementary Materials). After the crystal structure (I) determination (see below), a theoretical powder pattern (Figure S1b) was calculated, based on the .cif file using the STOE XPow program [5] with the indexing of reflections. The simulated pattern matches well with the experimentally registered one. The XRD pattern for the total mass of crystals in the experiment demonstrated a predominance of crystals (II)—known orthorhombic $Pb(IO_3)_2$—in relation to a smaller number of crystals (I); the experimental and theoretical patterns for $Pb(IO_3)_2$ are given on Figure S1c,d.

A second harmonic generation measurement was carried out on the crystalline samples, according to the Kurtz and Perry scheme [6]. A Minilite-I YAG:Nd laser in Q-switched mode at the repetition rate of 10 Hz was used as a source of radiation at wavelength $\lambda_\omega$ = 1.064 mcm. The measurements of light with a doubled frequency at 0.532 mcm were performed in the reflection mode, which allowed the exclusion of the influence of the powder thickness on the SHG output. A standard $\alpha$-$SiO_2$ powder sample of 5 mcm grain size was used as a reference. The powder samples for the SHG were prepared by grinding in a mortar of crystals (I) carefully selected from the total mass of the product of the hydrothermal synthesis. After grinding, the powders were separated with sieves into fractions of different grain sizes (30–100 mcm). The finest powder, of 5 mcm grains, was prepared by prolonged grinding of the crystals in alcohol until a suspension was formed, which was then dried. This fraction was used for a quantitative comparison of SHG activity with an $\alpha$-quartz powder standard with the same grain size.

The SHG output signal intensity, and its dependence on grain size (Figure 1), is typical for phase-matching materials that belong to class A, after Kurtz [6], and similar to

those previously observed for the iodates of $Rb_3Sc(IO_3)_6$ family [3]. The relative intensity, $I_{2\omega}/I_{2\omega}(SiO_2) = 10$, of the $PbF(IO_3)$ and quartz in powders with a grain size of $l = 5$ mcm is not influenced by phase-matching conditions, which work at $l \sim L_{coh}$ and $l > L_{coh}$. As far as $L_{coh} = 20$ mcm for $SiO_2$ and $L_{coh} \rightarrow \infty$ for $PbF(IO_3)$, a simplified relationship from [6] may be used in the region of $l << 20$ mcm in order to calculate the second-order optical nonlinearity for $PbF(IO_3)$:

$$<d_{eff}> \approx 0.365 \text{ pm/V}[I_{2\omega}/I_{2\omega} (SiO_2)]^{1/2} .$$

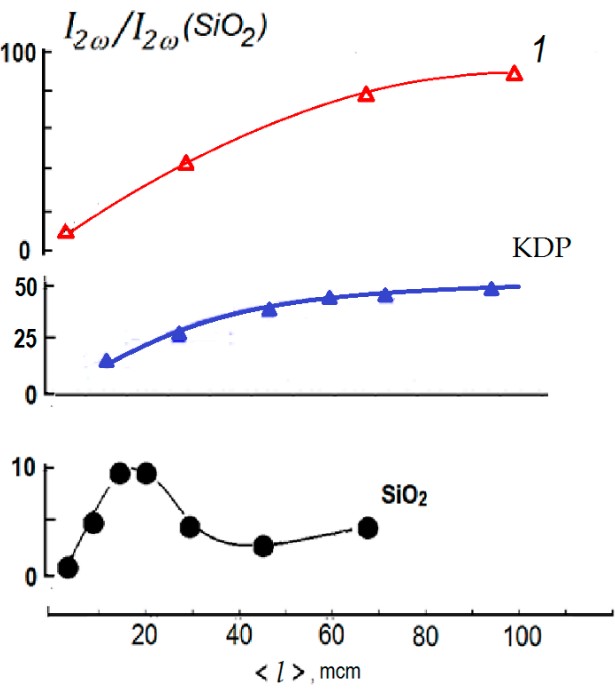

**Figure 1.** Powder SHG response of two experiments of $PbF(IO_3)$-1 in comparison with α-quartz and $KH_2PO_4$ depending on average grain size in the powder.

Thus, the space-averaged, effective second-order nonlinear optical coefficient $<d_{eff}> = 1.15$ pm/V. The SHG output from 100 mcm $PbF(IO_3)$ powder is approximately two times larger in relation to the analogous KDP ($KH_2PO_4$) powder, which is close to other metal iodate fluorides [1,2].

The unit cell of (I) was determined on a single crystal using an XCalibur S diffractometer equipped with a CCD area detector (ω scanning mode) and graphite-monochromatic Mo $K_\alpha$ radiation source (λ = 0.71073 Å). The pre-experiment shows two variants of unit cells: orthorhombic and monoclinic. A small, colorless transparent crystal with area dimensions of 0.11 × 0.10 × 0.03 mm was selected for the single-crystal X-ray study for the diffraction experiment, carried out on the same diffractometer at ambient conditions. The refined unit-cell parameters allowed indexing again with orthorhombic and monoclinic axes. The data for the two settings were integrated using the CrysAlis Pro Agilent Technologies software [7] and corrected for the Lorentz-polarization factor.

For clarifying the true symmetry and structure, a second experiment was carried out at 150 K using a Bruker AXS diffractometer equipped with a CCD area detector (ω scanning mode) and graphite-monochromatic Mo $K_\alpha$ radiation source (λ = 0.71073 Å) on a colorless transparent crystal with a size of 0.15 × 0.05 × 0.025 mm. The data were integrated, scaled, and merged using the SAINT program [8]. Based on the whole experimental data set, a monoclinic unit cell with a negligible deviation in two angles of 90° was registered and used in the following calculations.

### 2.2. Crystal Structure PbF(IO₃)

2.2.1. Orthorhombic Model, Ambient Conditions Data

The orthorhombic symmetry was tested with the unit cell $a$ = 6.0438(2), $b$ = 5.7840(2), $c$ = 11.0731(3) determined by the CrysAlis program. The unit cell dimensions correlate with the unit cell of BiO(IO₃) [9], $a$ = 5.658, $b$ = 11.039, $c$ = 5.748, which correspond to the *Pca*2₁ space group. The polar space group *C2ma* (*Abm*2 in the standard setting) was selected for the new lead fluoride iodate in contrast with the centrosymmetric *Cmma* suggested by the CrysAlis program (*C*-lattice was confirmed by the extinction of reflections), which was impossible because of the optical nonlinearity. The search of the model by direct methods in SHELXS [10] allowed us to find two heavy atoms: Pb and I, coordinated by O atoms, whose positions were found from the residual density with R~5%. The heavy atoms were on the mirror plane, which doubled the O atoms up to split umbrella-like IO₃⁻ groups with short O-O distances, which required half occupations of O positions and a disordered model. It was necessary to introduce an absorption correction, which was done using numerical Gaussian integration over a multifaceted crystal model [11]. The refinement of the model in SHELXL [10] gave satisfactory atomic displacement parameters, interatomic distances (except splitting), and R-factor. One of the O atoms was an F atom; based on the temperature displacement parameters, it was the atom at the center of the Pb tetrahedra (anion-centering polyhedra) presented in previously determined structures (see below) also by an F atom or by an (OH) group. The model was neutral, based on Pauling's balance of valences. The resulting chemical formula is PbF(IO₃), Z = 2.

The monoclinic cell, as a second variant of the indexing, was a = 4.1867(1), b = 11.0837(3), c = 4.1870(1), and β = 92.518(3) Å (mineralogical setting) and has not been previously defined. This unit cell correlates with the orthorhombic, with $c_{orth}$ equal to $b_{mon}$, and the choice of two orthorhombic C vectors as the $a,c$-axis of the new monoclinic cell (Figure 2).

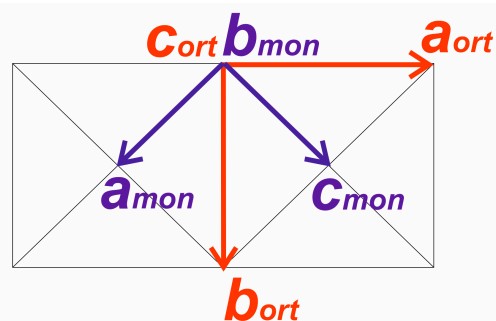

**Figure 2.** Monoclinic *P*-lattice (blue) selected from the orthorhombic *C*-lattice (red).

As before, the space group suggested by CrysAlis was P12/*n*1, and acentric P1*n*1 was used in the calculations. The next calculations for the monoclinic version of the structure were carried out using the low-temperature data, as the low-temperature experiment showed a definite monoclinic cell.

2.2.2. Monoclinic Model, Low-Temperature Data

The Mmonoclinic cell parameters at of 150 K a = 4.1581(4), b = 4.1548(4), c = 11.042(1) Å, γ = 92.470 (5) showed contraction because of cooling compared with the cell at ambient conditions. The Ppolar acentric space group P11n (standard setting) was consistent with a strong SHG signal, extinctions, and unit cell angless. A Ssemi-empirical absorption correction was thean applied, based on symmetrically equivalent reflections in the SADABS program [8]. The Ddirect methods in SHELXS [10] gave, again, two positions for the heavy atoms Pb and I, and a series of O⁻ atoms obtained in the difference Fourier synthesis on the distances typical for the (IO₃)⁻ group. As before in the orthorhombic model, the doubling of O⁻ atoms in the umbrella-like IO₃⁻ groups appeared and required an the explanation. It was supposed that the mirror plane is a twin element in a monoclinic group, which leads to overestimatoverestimating theed symmetry as orthorhombic (see room temperature

experiment). Three $O^-$ atoms forming the normal umbrella coordination of I -atoms were selected and refined in SHELXL [10]. Twinning was introduced by matrix 010/100/001 for the diagonal vertical mirror plane, which reduced the R-factor. The components of the twin were refined, BASF = 0.58, and the additional peaks, correspondinged to the doubled by twinning $O^-$ atoms, disappeared from the residual electron density. The final formula was the same $PbF(IO_3)$, Z = 2. The structural model was finally refined using the least squares procedure in an anisotropic approximation for the atomic displacements for all the atoms, correction of the anomalous scattering, and with the refinement of the weighting scheme using SHELXL [10]. The Ssubstitution of O for F lowered the R-factor and confirmed the presence of the F -atom by the anisotropic displacement parameters. The true structure of the new modification is monoclinic with twinning by the mirror plane. If this mirror plane is considered as a real symmetry element, the structure of the merohedral twin is described as orthorhombic C2ma and corresponds to the structure determined on the base of the ambient conditions data.

The crystallographic data, atomic coordinates, and selected bonds are presented in Table 1, Table S1–S4. CCDC (ICSD) 2212814 and 2212815 contain, accordingly, the crystallographic data for the orthorhombic average structure and monoclinic true structure, refined with a twin component. These data can be obtained free of charge via www.ccdc.cam.ac.uk/data_request/cif (accessed on 13 October 2022). All the illustrations were produced using the ATOMS [12] and CORELDRAW programs.

**Table 1.** Crystal data and structure refinement for $PbF(IO_3)$, monoclinic and orthorhombic unit cells.

| Formula | $PbF(IO_3)$ | |
|---|---|---|
| Formula weight | 401.09 | |
| Crystal system | monoclinic | orthorhombic |
| Space group, Z | *Pn*, 2 | *C2ma*, 4 |
| $a$, Å | 4.1581(4) | 6.0438(1) |
| $b$, Å, | 4.1548(4) | 5.7840(1) |
| $c$, Å, $\gamma$, ° | 11.0416(10), 92.470(5) | 11.0731(3) |
| V, Å$^3$ | 190.58(3) | 387.085(2) |
| $D_x$, r/см$^3$ | 6.990 | 6.882 |
| $\mu$, мм$^{-1}$ | 52.240 | 51.440 |
| Wavelength, Å | 0.71073 | 0.71073 |
| $\Theta$ range/degree | 3.69–38.9 | 3.68–30.63 |
| Refl. collected/unique/$R_{int.}$ | 3613/1755/0.0651 | 3000/615/0.0871 |
| Completeness to $\Theta$, % | 0.927 | 0.966 |
| Parameters | 56 | 28 |
| GOF (S) | 1.115 | 1.104 |
| $R_{all}$, $R_{gt}$, $R_{wgt}$ | 0.0522, 0.0504, 0.1314 | 0.0471, 0.0431, 0.1025 |
| $\Delta\rho_{min}/\Delta\rho_{max}$, э/Å$^3$ | −14.387/3.760 | −5.324/2.228 |

Note: $R_{gt} = \Sigma. \; ||F_o| - |F_c||/ = \Sigma|F_o|$ and $R_{wgt} = [\Sigma \, w \, (F_o^2 - F_c^2)^2 / \Sigma \, w \, (F_o^2)^2]^{1/2}$ for $F_o^2 > 2\sigma \, (F_o^2)$.

## 3. Crystal Chemistry of Series of Related Structures

### 3.1. Some Information on Aurivillius-like Compounds

Iodate fluoride $PbF(IO_3)$ has many important structural features common with the Aurivillius phases. The interest in Aurivillius-like, n = 1, oxyfluorides such as $Bi_2TiO_4F_2$ or $Bi_2CoO_2F_4$ over the last few years is due to their properties (see papers [13,14] and numerous references therein). For the first compound, the key structural parts are Ti and Co octahedra, which form perovskite-like layers between the $Bi_2O_2$ fluorite-like layers. The first oxyfluorides synthesized by solid-state reaction and investigated by a Rietveld analysis possessed photocatalytic and dielectric behaviors, depending sensitively on their crystal structure and symmetry. The role of these factors led the authors to the conclusion that the anion order plays an important role and may tune the symmetry up to breaking the inversion center. The second compound has been a research focus due to its complexity of magnetic properties. It was synthesized hydrothermally. A combination of the Rietveld

method, SHG measurements, magnetic structure determination on high-resolution NPD data, and theoretical calculations led to an explanation of its multiferroic and magnetoelectric properties.

Many substances characterized as Aurivillius-like phases have variations in the fluorite-like layer. Some of them attract attention due to their water-splitting photocatalytic properties, for example, a series of compounds, $Bi_4AO_6Cl_2$ (A = Ba, Sr, Ca) with "double" and "triple" fluorite layers [15], which correctly correspond to the single and double case. The authors discuss the structures with the layers presented by $(PbBiO_2)^+{}_{\infty\infty}$ (one fluorite-like layer) in $PbBiO_2Cl$, $(YBi_2O_4)^+{}_{\infty\infty}$ (doubled fluorite-like layer) in $Bi_2YO_4Cl$, and both types presented simultaneously in $Bi_4BaO_6Cl_2$ as $(Bi_2O_2)^{2+}{}_{\infty\infty} + (BaBi_2O_4)_{\infty\infty}$, (Figure 3). They alternate with the single layers of Cl atoms instead of octahedral layers (perovskite-like) in the structures discussed above.

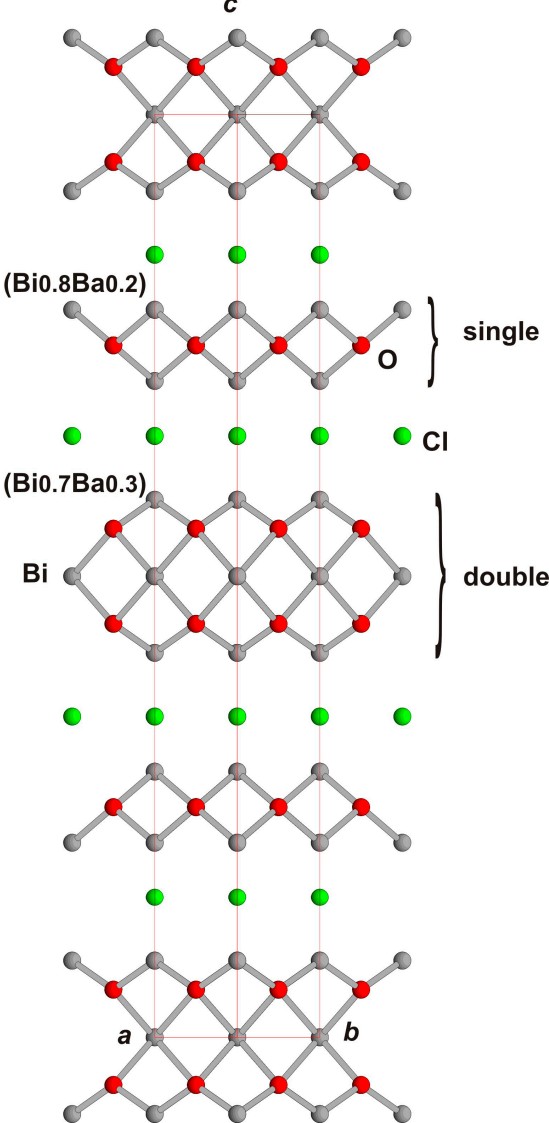

**Figure 3.** $Bi_4BaO_6Cl_2$ crystal structure with single and double fluorite layers and single Cl layer. Single and double fluorite-like layers are marked by brackets. Ball-and-stick presentation is used here and after. Bi/Ba, O, and Cl ions are shown in grey, red, and green, respectively. Axes notations *a, b, c* are given here and after.

The chlorine layers may be presented also by other anionic units, such as iodate or bromate groups, as discussed in [16]. The comparison of BiOCl, a bismoclite structure in

common with BiOX, X = Br, I, and BiOXO$_3$, X = Br, I, was carried out in [16]. The compound BiOBrO$_3$ was obtained for the first time in [16]. The fluorite layers alternate with double chlorine layers or BrO$_3$ layers (Figure 4a,b). The investigation by a series of methods allowed the derivation of a sequence of compounds, BiOBrO$_3$ > BiOI > BiOIO$_3$ > BiOBr, in relation to the effectiveness of photocatalytic properties.

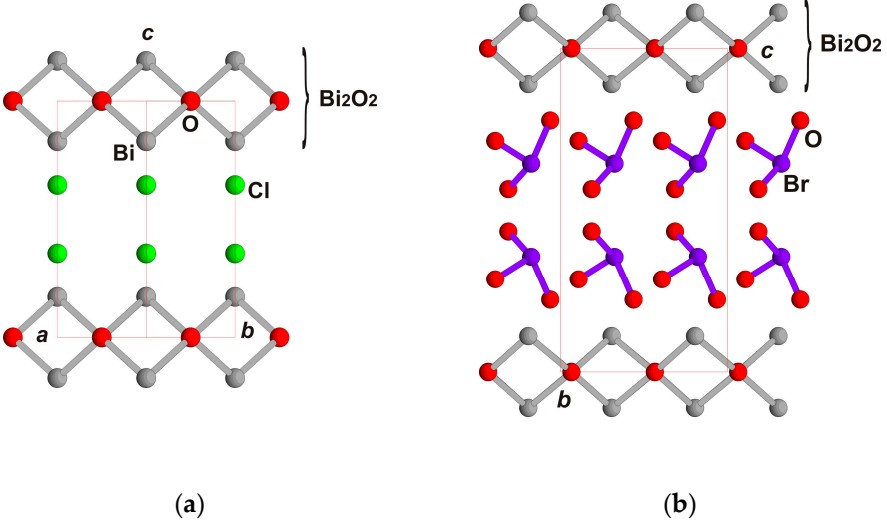

(**a**)　　　　　　　　　　　　(**b**)

**Figure 4.** (**a**) BiOCl structure with single fluorite layer and double Cl layer. Bi, O, and Cl ions are shown in grey, red, and green, respectively. (**b**) BiO(BrO$_3$) structure with single fluorite layer and double BrO$_3$ layer instead of Cl double layer. Bi, Br, and O ions are shown in grey, purple, and red, respectively.

*3.2. New Monoclinic PbF(IO$_3$) and Its Comparison with the Earlier Known Modification of PbF(IO$_3$), BiO(IO$_3$), and Other Members of the Proposed Family; Structure–Property Relations*

The structure of the new monoclinic modification of PbF(IO$_3$), space group $P11n$, is of the layered type (Figure 5a,b). One of the layers, (PbF)$^+_{\infty\infty}$, is similar to the neutral (PbO)$_{\infty\infty}$ layer in the mineral PbO (lithargite), a = 4.004, c = 5.071 Å, with an *a,b*-axis close to ours and a *c*-axis two times smaller. This layer designation as the "fluorite layer" (separated from the fluorite structure) is more common. Such layers were described as anion-centered tetrahedral layers [Ba(OH)]$^+_{\infty\infty}$ in Ba(OH)(IO$_3$), space group *Cm* [17], (Figure 6a). The local symmetry of the layer determined there corresponds to the symmetry of lithargite $P4/nmm$. The layers (BiO)$^+_{\infty\infty}$ were first described for BiO(IO$_3$), space group $Pca2_1$, as a fragment of " . . . Aurivillius phases, however, instead of perovskite-like anion blocks separating the layers; locally, polar iodate anions are observed." [9] (Figure 6b). This description characterizes the main feature of the new, selected structural family. Fragments of lithargite or fluorite structures are equal in terms of symmetrical and topological peculiarities.

The previously determined orthorhombic modification of PbF(IO$_3$), space group *Iba*2 [18], is composed of (PbF)$^+_{\infty\infty}$ layers, equal to the layers in the new, monoclinic one, which alternate with the (IO$_3$)$^-$ "layers" (Figure 6c). The interatomic distances for both modifications are practically equal for Pb–F and I–O. The umbrella-like groups in all the compounds form a "layer" connected to the PbF, BiO, or BaOH layers above and below through common vertices, thus forming a sheet. Here, we can see an analogy with layered silicates, such as mica or talc minerals, with a central octahedral part and, attached from both sides (or one), tetrahedral parts in structures of trigonal symmetry. The tetragonal symmetry of the central part of the sheets is the common feature for all these structures. The alternation of sheets is distinctive for all of them, also. Let us compare both modifications of PbF(IO$_3$) in an orthorhombic setting. One independent sheet in the new modification of PbF(IO$_3$) is multiplied by the *c*-axis translation (*C*-lattice translation is in the sheet), (Figure 6d). In contrast, there are two independent sheets in the polytype studied

earlier [18], which are multiplied by the *I*-translation of the space group, (Figure 6c). This determines the unit cell parameters 11.07 Å in the new polytype compared to 22.14 Å in the previously studied. As shown before, the orthorhombic and monoclinic unit cells of the new modification transform one into another keeping the *c*-axis unchanged (Figure 2). They correspond to two settings of the unit cell of the same compound—the new modification of PbF(IO$_3$). The cleavage is typical for the compounds because the bonds between the sheets are weak, with only some interaction between the I atom from one sheet and the O4 from the next sheet at an enlarged distance.

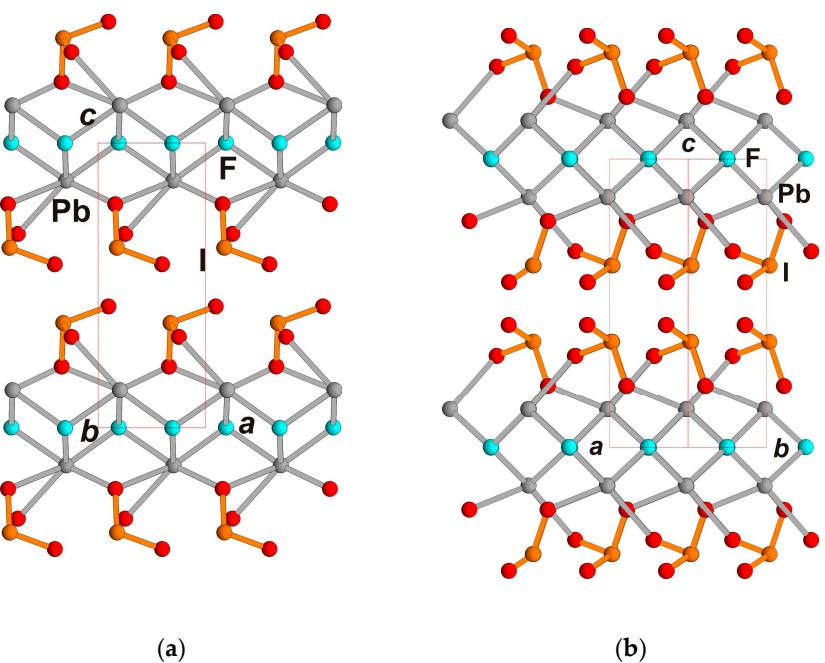

(**a**)                     (**b**)

**Figure 5.** PbF(IO$_3$) crystal structure. (**a**) *ac* projection, (**b**) diagonal *ab*-projection. Pb, I, F, and O ions are shown in grey, orange, blue, and red, respectively.

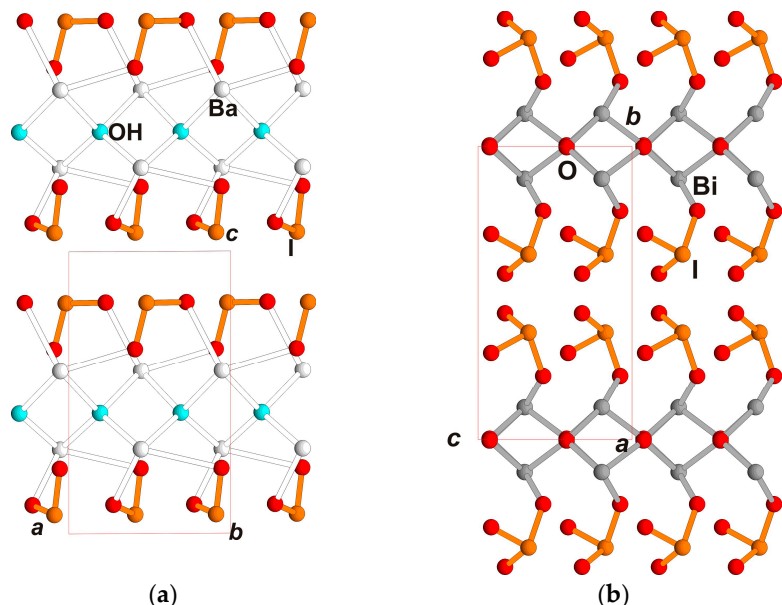

(**a**)                     (**b**)

**Figure 6.** *Cont*.

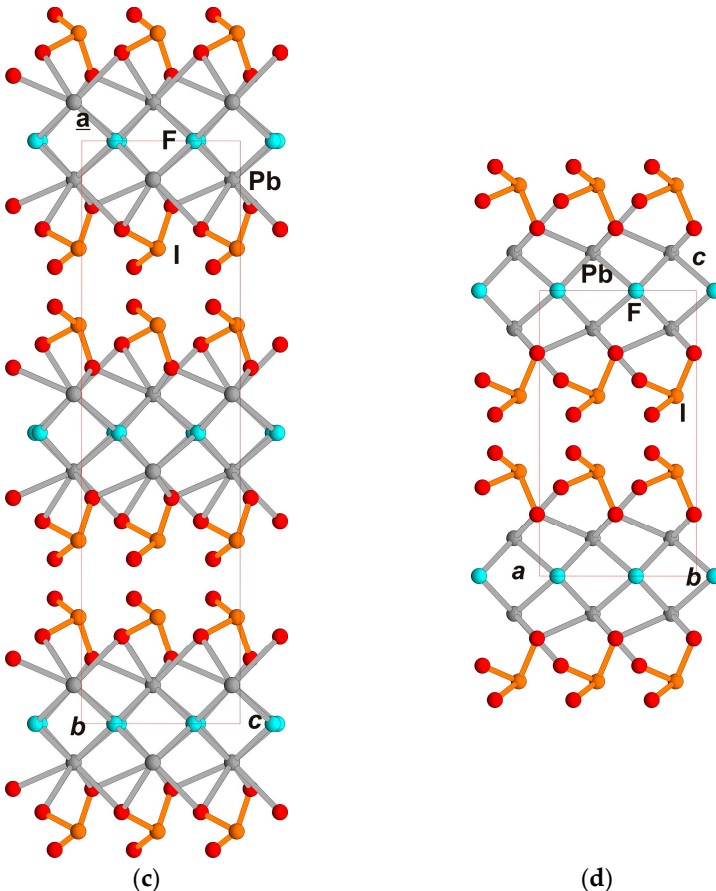

**Figure 6.** Projections of crystal structures in similar views. (**a**) Ba(OH)(IO$_3$) *ac* projection, (**b**) BiO(IO$_3$) *bc* projection, (**c**) PbF(IO$_3$) orthorhombic modification, *ac* projection, (**d**) PbF(IO$_3$) monoclinic modification in orthorhombic unit cell. Ba, Pb, I, F/OH, and O ions are shown in white, grey, orange, blue, and red, respectively.

The chemical formulae of the considered compounds are very similar and, taking into account the hetero-valence substitutions, may be written as MX(IO$_3$) with M = Bi$^{3+}$, Ba$^{2+}$, Pb$^{2+}$, X = O$^{2-}$ F$^-$, (OH)$^-$. It is obvious that they form a common family with the nonlinear optical properties of members with Bi$^{3+}$ and Pb$^{2+}$, having lone pairs. They possess high optical nonlinearity, determined mainly by the iodate groups as was discussed in the cited publications. The Ba(OH)(IO$_3$) compound has weak optical nonlinearity because of a practically anti-parallel arrangement of IO$_3$- groups around the central tetragonal layer. As a result, the sheets are very close to being centrosymmetric. For the recently obtained and investigated BaF(IO$_3$) [19], only the centrosymmetric modification similar to hydroxide iodate Ba(OH)(IO$_3$) [17] is realized.

## 4. Iodate Fluoride Compounds with Similar Structural Fragments

There are some substances whose structures are slightly different from those discussed above. In Pb(IO$_3$)$_2$, space group *Pbna* (standard setting *Pbcn*), no central tetragonal fluorite layer is presented, but the Pb polyhedral layer is in the same place [20] (Figure 7a). It was mentioned above that centrosymmetric Pb(IO$_3$)$_2$ was obtained in our experiment simultaneously with PbFIO$_3$. Another iodate compound is LiMoO$_3$(IO$_3$) [21]. The unit cells of both iodates have comparable *b,c*- (Figure 7a) and *a,b*-axes (Figure 7b) in the layer and different *a*- and *c*-vertical axes: the smaller one (*c*) 9 Å in the Mo iodate, and the larger one (*a*, doubled) 16.6 Å in the Pb iodate. The octahedral sheet in Mo iodate, which is in the fluorite layer position, has local symmetry *Pman* but the multiplication of iodate groups is produced only by 2$_1$ axes, which results in space group *P*2$_1$ (Figure 7b). Large Pb

polyhedra in $Pb(IO_3)_2$ have four short bonds (~2.5 Å) and four longer bonds (~2.8 Å). The arrangement of the iodate groups around the Pb is centrosymmetric. Despite the polarity of the pair of iodate layers, their multiplication in the structure is due to the inversion center. Therefore, the lattice parameter along the alternate layers in $Pb(IO_3)_2$ is doubled in comparison with $LiMoO_3(IO_3)$ (Figure 7a,b). However, this leaves $Pb(IO_3)_2$ a close relative to the family.

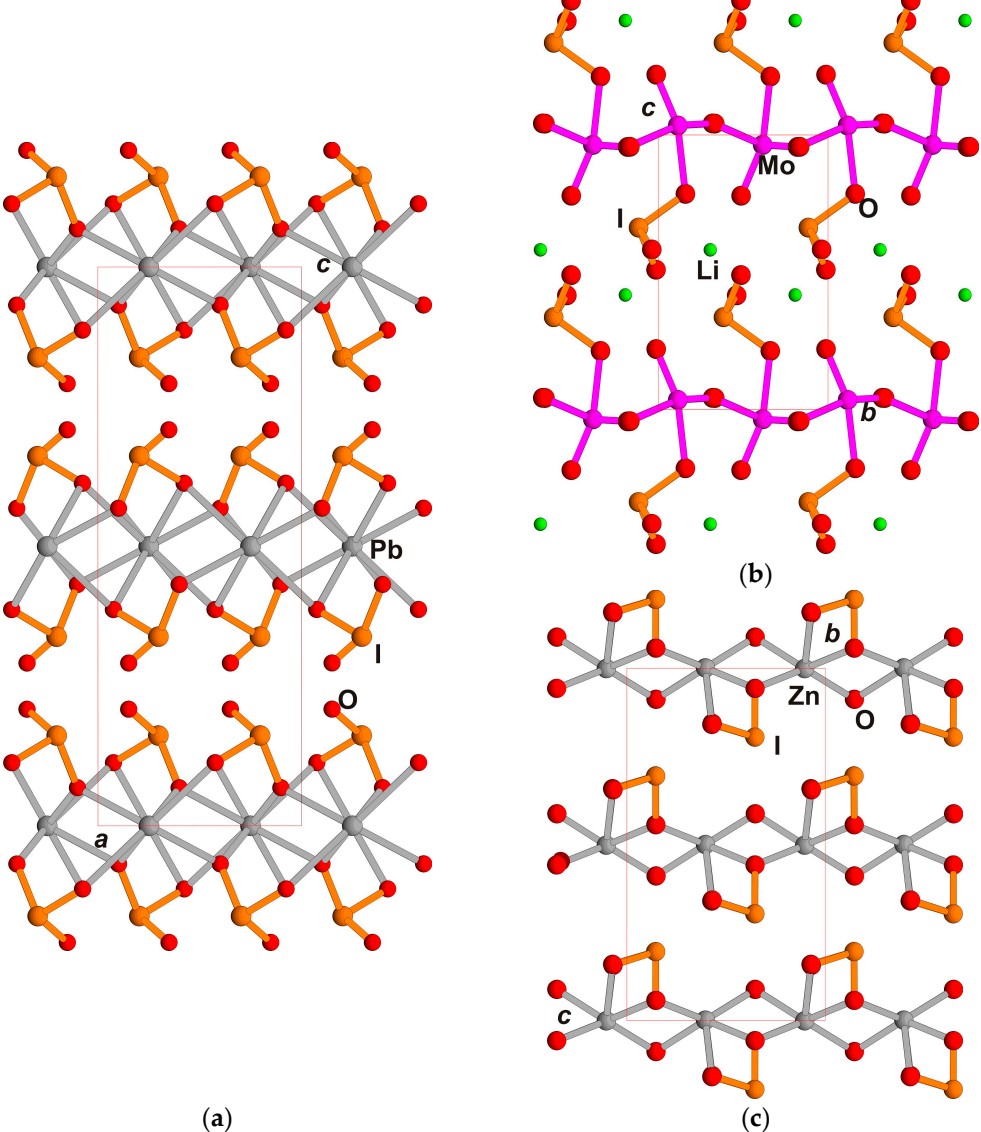

(a)

(b)

(c)

**Figure 7.** (**a**–**c**). Projections of crystal structures in similar views. (**a**) $Pb(IO_3)_2$ *ac* projection, (**b**) $LiMoO_3(IO_3)$ *bc* projection, (**c**) $ZnIO_3(OH)$ *bc* projection. Pb or Zn, I, Mo, Li, and O ions are shown in grey, orange, purple, green, and red, respectively.

New iodate $Zn(OH)IO_3$ [22] demonstrates a similarity in structure, despite its Zn atom being smaller compared to Pb and Bi and having a reduced coordination number, CN = 5. Zn atoms are located here in semi-octahedra, which are connected in chains and completed to layers by $IO_3$ groups (Figure 7c). This iodate is polar, possesses high optical nonlinearity, and is close in structure and properties to $LiMoO_3(IO_3)$.

Another new iodate with a similar formula is magnetic $MIO_3F$ (M = Co,Ni) [23]. The M polyhedra here form zigzag chains surrounded by $IO_3$- groups and then unite into a framework (Figure 8). The zigzag topology prevents the formation of layers, as in all the previous structures.

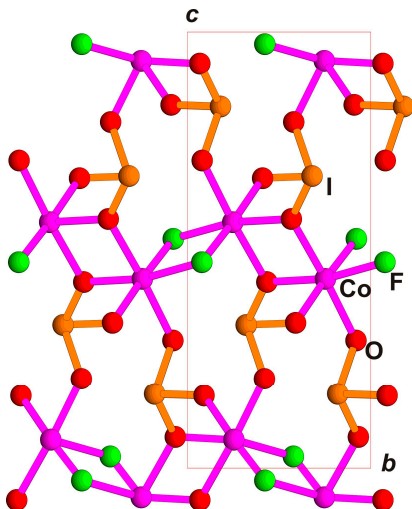

**Figure 8.** Crystal structure Co(IO$_3$)F in *bc* projection. Co, I, F, and O ions are shown in purple, orange, green, and red, respectively.

Iodate fluoride (NH$_4$)Bi$_2$(IO$_3$)$_2$F$_5$ [24] differs in chemical content from other iodates under consideration, but, nevertheless, it possesses some common features in its structure. The latter is composed of "one-side" sheets [Bi$_2$(IO$_3$)$_2$F$_5$]$^-$ or [Bi(IO$_3$)F$_{2.5}$]$^-$, which alternate with NH$_4$ groups (Figure 9). As in Pb(IO$_3$)$_2$, the coordination number of the largest cation (Bi$^{3+}$) in this compound is CN = 8. The structure may be presented as an alternation of iodate layers with the "layers" of NH$_4$ groups substituting for the iodate groups. With this exception, the structure of (NH$_4$)Bi$_2$(IO$_3$)$_2$F$_5$ has common features with the other iodates considered.

Despite the formula similarity, the efficient optical nonlinear crystal Bi(IO$_3$)F$_2$ [25] does not belong to the family of Aurivillius-type iodates. The layered structure of this crystal is represented by a polar [BiF$_2$]$^+$ framework with tunnels. The tunnels are filled with (IO$_3$)$^-$ groups, interconnected in the framework by large cationic polyhedra.

The structural data for the analyzed compounds are presented in Table 2.

**Table 2.** Structural data for the analyzed compounds.

| Compound | Space Group | *a*, Å | *b*, Å, β, ° | *c*, Å, γ, ° | Type of Sheets | Reference |
|---|---|---|---|---|---|---|
| | | | Family AX(IO$_3$), A = Bi, Ba, Pb, X= O, F, (OH) | | | |
| BiO(IO$_3$) | *Pca*2$_1$ | 5.6584 | 11.0386 | 5.7476 | | [19] |
| BiO(BrO$_3$) | *Pca*2$_1$ | 5.647 | 10.926 | 5.614 | | [16] |
| PbF(IO$_3$) | *Iba*2 | 22.140 | 5.786 | 6.045 | Single fluorite-like + | [18] |
| PbF(IO$_3$) | *P*11*n* | 4.1581 | 4.1548 | 11.0416, 92.47 | iodate (bromate) instead of perovskite | Present work |
| Ba(OH)(IO$_3$) | *Cm* | 6.0582 | 6.3509, 90.338 | 10.5825 | | [17] |
| BaF(IO$_3$) | *P*2$_1$/*c* | 10.5729 | 6.3354, 90.497 | 6.045 | | [19] |
| | | | Related compounds | | | |
| Bi$_4$AO$_6$Cl$_2$ (A = Ba, Sr, Ca) | *I*4/*mmm* | 4.0028 | | 31.563 | Single and double fluorite-like + single chloride | [15] |
| BiOCl | *P*4/*nmm* | 3.8870 | | 7.3540 | Single fluorite-like + double chloride | [16] |
| Pb(IO$_3$)$_2$ | *Pbna* | 5.558 | 6.040 | 16.650 | Octahedral instead of fluorite-like + double iodate instead of chloride (perovskite) | [20] |
| LiMoO$_3$(IO$_3$) | *P*2$_1$ | 5.4104 | 5.3158, 106.87 | 9.0025 | Octahedral instead of fluorite-like + double iodate instead of chloride (perovskite) with Li atoms | [21] |
| Zn(OH)IO$_3$ | *Cc* | 4.6767 | 11.2392, 90.02 | 6.3308 | Semi-octahedral instead of fluorite-like + double iodate | [22] |
| MIO$_3$F (M = Co,Ni) | *P*2$_1$/*n* | 4.9954 | 5.211, 95.35 | 12.5179 | Zigzag octahedral instead of fluorite-like + single iodate | [23] |
| (NH$_4$)Bi$_2$(IO$_3$)$_2$F$_5$ | *P*2$_1$ | 5.718 | 5.892, 100.38 | 15.176 | Polyhedral Pb-layer + double iodate alternating with single NH$_4$ | [24] |
| Bi(IO$_3$)F$_2$ | *C*2 | 12.8275 | 5.3089, 101.35 | 6.0790 | Framework structure | [25] |

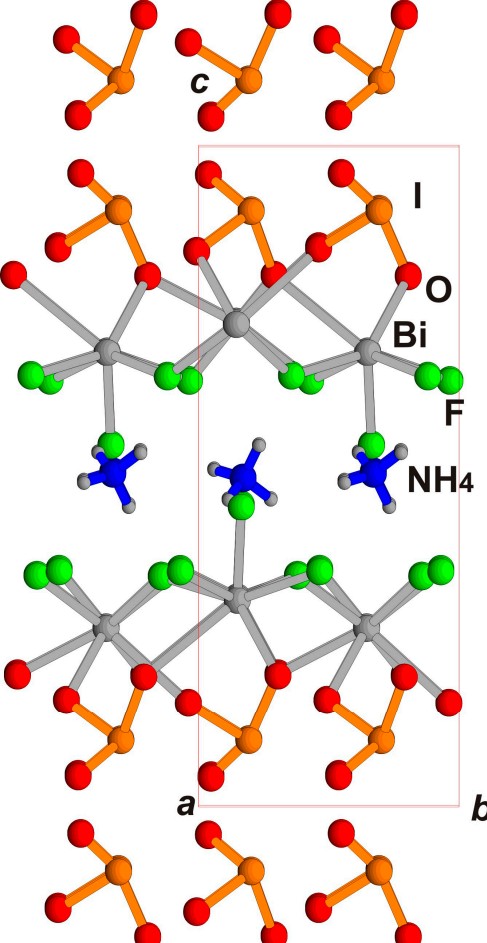

**Figure 9.** Crystal structure of $(NH_4)Bi_2(IO_3)_2F_5$ in *bc* projection Bi, I, N, H, and O ions are shown in grey, orange, blue, small grey, and red, respectively.

## 5. Conclusions

New crystals of $PbF(IO_3)$ polytype modification are synthesized hydrothermally and demonstrate a strong SHG optical response. They are phase-matchable at the fundamental wavelength of 1064 nm and possess SHG intensities around two times those of $KH_2PO_4$ (KDP). The crystal structure was solved in two space groups, the orthorhombic *C2ma* and the monoclinic *Pn*, of which the monoclinic is true and is described with a twinning by a mirror plane introduced in the structural refinements. The orthorhombic variant presents a twin superstructure with a mirror plane as the element of the space group. The orthorhombic symmetry used in the comparison with the related structures and deviation showed close similarities in the selected family $MX(IO_3)$, M = Bi, Ba, Pb, X = O, F, (OH) with a series of members: orthorhombic $PbF(IO_3)$, monoclinic $PbF(IO_3)$, $BiO(IO_3)$, $Ba(OH)(IO_3)$, and $BaF(IO_3)$. All of them have a polytypic nature with sheets composed of a central pseudo-tetragonal anion-centered layer $(MX)_{\infty\infty}$, similar to the layer in the PbO (lithargite) structure and attached from both its side $(IO_3)$ groups. The central layer is also described as a fluorite-like layer. These compounds were also characterized as similar to Aurivillius phases with fluorite-like layers and perovskite-like layers substituted by $(IO_3)$ groups.

Our new crystals demonstrate a disorder presented as twinning. It is proven in our experiment that we obtained the simplest structure in the whole polytype family, one which has the smallest unit cell and the lowest symmetry in contrast with the previously determined polytype and other members of the family. Let us analyze the reason for the formation of different polytypes. Despite the fact that the new modification of $PbF(IO_3)$ was synthesized under hydrothermal conditions, and the previously known one was



obtained during unconventional low-temperature solid-phase technology, the conditions for obtaining both phases are similar. Both syntheses proceeded at a temperature of 270 °C in autoclaves of approximately equal capacity. $HIO_3$ and $NH_4NO_3$, which were used to obtain the previous modification, decompose at a temperature of about 200 °C to form water; thus, the reaction proceeded in an aqueous solution. The main difference in the formation of the two polytypes is the reaction rate and the amount of solvent. The synthesis of the phase described in this article proceeds with a large amount of solvent at a lower reaction rate, which contributed to the formation of the simpler polytype structure. The growth conditions of previously obtained $PbF(IO_3)$ modification favor a synthesis to the most complicated polytype in the family.

The optical nonlinearity of the iodates of the Aurivillius family and structurally related iodates is determined by the polar orientation of the iodate groups, which make an overwhelming contribution to the optical nonlinearity. From the crystal chemistry point of view, the heavy atoms in these structures are located in the second cation environments in relation to the iodate groups and indirectly affect the nonlinearity. In particular, large Ba cations without single electron pairs provoke a symmetric variant of the Aurivillius-type structure, in contrast to the acentric $Bi^{3+}$ and $Pb^{2+}$ cations known in polar iodates with strong second-order optical nonlinearity.

There is wide diversity in the extended series of related compounds, which includes variations of fluorite-like layers (single or double), perovskite-like layers presented by octahedral or more complicated polyhedral by $IO_3$ ($BrO_3$) groups, Cl atoms, or $NH_4$-groups. This allows the development of future search for new promising phases.

**Supplementary Materials:** The following supporting information can be downloaded at: https://www.mdpi.com/article/10.3390/sym15010100/s1, Figure S1: Experimental powder diffraction pattern of $PbF(IO_3)$ crystals, DRON-UM diffractometer, Cu K$\alpha$ radiation, 2$\theta$-interval 6–70.0°, Table S1: Atomic coordinates and equivalent displacement parameters for $PbF(IO_3)$ orthorhombic unit cell. Uequ is defined as one third of the trace of the orthogonalized Uij tensor, Table S2: Interatomic distances for $PbF(IO_3)$ orthorhombic unit cell, Table S3: Atomic coordinates and equivalent displacement parameters for $PbF(IO_3)$ monoclinic unit cell. Uequ is defined as one third of the trace of the orthogonalized Uij tensor, Table S4: Interatomic distances for $PbF(IO_3)$ monoclinic unit cell.

**Author Contributions:** Conceptualization, E.B. and O.D.; methodology E.B., S.S. and O.D.; software O.R. and E.B.; investigation O.R., A.V. and S.S.; resources O.D., A.V., S.S. and E.B.; writing—original draft preparation O.R., E.B., O.D., A.V. and S.S.; writing—review and editing, E.B. and S.S.; visualization O.R. and E.B. All authors have read and agreed to the published version of the manuscript.

**Funding:** This research received no external funding.

**Institutional Review Board Statement:** Not applicable.

**Informed Consent Statement:** Not applicable.

**Data Availability Statement:** CCDC (ICSD) 2186415 and 2186468 contain crystallographic data for this paper. These data can be obtained free of charge via www.ccdc.cam.ac.uk/data_request/cif (accessed on 13 October 2022).

**Acknowledgments:** The authors are grateful to Natalie Zubkova for her aid in collection of experimental diffraction data at ambient conditions using XCalibur diffractometer, geological faculty Lomonosov MSU; to Yulia Nelyubina for her aid in collection of experimental diffraction data at low temperature using Bruker diffractometer, Center for molecular composition studies, Nesmeyanov INEOS; and to Vasilij Yapaskurt for determination of compositions, the Laboratory of local methods of materials investigation, geological faculty, Lomonosov MSU.

**Conflicts of Interest:** The authors declare no conflict of interest.

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
