# Peer review of "New Modification of Polar Nonlinear Optical Iodate Fluoride PbF(IO3), the Family MX(IO3), M = Bi, Ba, Pb, X = O, F, (OH) Related to Aurivillius Phases and Similar Iodates"

_symmetry, doi:10.3390/sym15010100_

Round 1

Reviewer 1 Report

The authors reported the manuscript entitled “New modification of polar nonlinear optical iodate fluoride PbF(IO3), the family MX(IO3), M = Bi, Ba, Pb, X= O, F, (OH) related to Aurivillius phases and similar iodates”. The authors mainly analyzed the characteristics of the structures. However, the optical and thermodynamic properties have not been reported fully. So, I recommended this manuscript to be published after minor revision.

 1 The authors carried out the single crystal X-ray diffractometer method, the cif and checkcif reported should be provided, and the space group should be rechecked for I couldn’t find the space group Pn in the 230 space group in the table of Crystallographic Space Groups

 2 In the last paragraph of the first part (line 56), the sentence “in this review, …” made me confused about the type of this manuscript. Whether is the communication, article or review? The authors should check the manuscript carefully to avoid these confusions.

3 The legends of the Figure 3-8 should be given.

4 Other optical or thermodynamic properties of the PbF(IO3) could be provided, such as the absorption edge or TG and DTA curves.

Author Response

We are very thankful to the Reviewer 1 for reading of our manuscript with the attention and for valuable remarks.

  1. This group P11n corresponds to nonstandard setting of monoclinic group №7 in the International tables of crystallography. If the monoclinic axis is c-axis as in our case it is P11c in standard setting. Choosing of diagonal direction a,b-as unit cell parameter  with the keeping of previous c-axis for our structure will give standard setting  P11c. Using of nonstandard P11n, or P112/n is enough widespread and depends of correlation between axes dimensions.
  2. Reviewer rightly drew attention to the somewhat unusual format of our article, which is not formally a review, but contains significant review material on structural features of nonlinear optical iodates close to Aurivillius phases. Namely, we have formed a family of 6 compounds and another 9 compounds closely related and having similar fragments. The family is illustrated by a large number of figures, which determines the review nature of the work. The article is focused on solving a fundamental issue on the general principles of crystal chemistry of an important group of iodates, the main of which is the possibility of deviation of their symmetry from the maximum.
  3. In agreement with the Reviewer’s remark, we added legends in Figures captions. Also we made clearer our Figure 3.
  4. The details of optical properties, such as the optical indicatrix, the exact value of the phase-matching angle, the shape and position of the intrinsic absorption edge, and many others are usually determined in the framework of materials science researches, to which our study does not apply. In our work, the issues of optical materials science could not be raised due to the limited volume of the material obtained and the general focus of the work on solving a specific structural problem, mainly by the methods of X-ray diffraction experiment. The thermometric and optical data mentioned by the Reviewer will probably be used in the future in the study of larger batches of these substances and at presence of larger crystals. We see our contribution to the development of nonlinear optical iodates in determining the structure of a new PbF(IO3) modification, studying the conditions for obtaining crystals of a given structure, as well as crystal chemical analysis of their various varieties, which is noted in our article

Reviewer 2 Report

Review of the manuscript Symmetry- 2118424

New modification of polar nonlinear optical iodate fluoride PbF(IO3), the family MX(IO3), M=Bi, Ba, Pb, X=O,F, (OH) related to Aurivillius phases and similar iodates

By Elena Belokoneva, Olga Reutova, Anatoly Volkov, Olga Dimitrova and Sergey Stefanovich.

The authors obtained and investigated experimentally a novel non-centrosymmetric single crystal PbF(IO3) characterized by a large second-order optical nonlinearity. It has been shown experimentally that the new crystals demonstrated strong second   harmonic generation (SHG) effect.  The parameters of the PbF(IO3) crystal structure have been measured. It has been shown that iodate fluoride PbF(IO3) has common structural features with a number of compounds such as BiO(IO3), BiO(BrO3), Ba(OH) (IO3), BaF(IO3), Bi4AO6Cl2 (A=Ba, Sr, Ca), BiOCl, Pb(IO3)2, LiMoO3(IO3), Zn(OH) IO3, MIO3F, MIO3F (M=Co, Ni), (NH4)Bi2(IO3)2F5, Bi(IO3)F2. The structure characteristics of PbF(IO3) and of these compounds are presented in Table 2 (pp. 12-14) for a comparison. The authors also carried out the numerical simulations of the crystal structure. The numerical simulation results and the experimental results appeared to be in a good agreement.

 The synthesis of the new modification of PbF(IO3) is briefly described. The authors have shown that the strong second-order optical nonlinearity of the iodates of the Aurivillius family and structurally related iodates is caused by polar orientation of the iodate groups.

The paper contains interesting novel experimental results. It is well organized and clearly written.

The paper may be interesting for the researchers occupied in the physics and chemistry of crystals, nonlinear optics and spectroscopy.

The manuscript can be accepted for publication in a present form.

Author Response

We are very thankful to the Reviewer for careful reading of our manuscript, expressed interest and understanding for the content of the article.

Reviewer 3 Report

The authors describe a new phase of PbF(IO3)  showing SGH optics related to Aurivillius phases.

The article is well structured and clear to follow, the English should however be thoroughly checked in particular with respect to the use of articles. (see also further below)

The only technical/scientific issue I have is related to the structure of the monoclinic phase of PbF(IO3) reported here:

Cell: a 4.1581(4)Å b 4.1548(4)Å c 11.0416(10)Å, α 90.114(5)° β 89.976(5)° γ 92.470(5)°

For a monoclinic structure 2 angles should be 90°, the fact that an alternative setting of axis is used (with c axis as principal axis) if perfectly fine, as it allows a more easy comparison to the other phase. The refinement NEEDS to be repeated with a constrained unit cell (alpha and beta = 90° in this case)

the abstract could be modified as:

Abstract: A new modification of PbF(IO3) has been obtained as single crystals from hydrothermal synthesis, alongside the known centrosymmetric Pb(IO3)2 as a second phase. Measured with the Kurtz-Perry SHG method the new crystals are phase-matchable for YAG:Nd laser radiation and demonstrate strong SHG output. According to Xray-diffraction analysis fulfilled on a single crystal at low temperature the new crystals appeared monoclinic, space group Pn, opposed to the known orthorhombic modification of PbF(IO3), space group Iba2. The new crystals were also measured at room temperature showing an orthorhombic disordered variant of the new phase (space group C2ma, standard Abm2). This variant presents an “averaged structure” with a mirror plane in the group. The low temperature X-ray single crystal experiment allowed to find correct structural model where the mirror plane was found as a twin element in the real monoclinic Pn structure. Careful crystal chemical analysis led to a whole family of non-linear optical crystals with the common formula AX(IO3), A = Bi, Ba, Pb, X= O, F, (OH) currently counting 6 representatives including well the known BiO(IO3). All of them possess common central cationic layers similar to the known Aurivillius-type phases with anionic iodate layers attached above and below these layers instead of perovskite-like, or halogens. Structure-properties relations are discussed with respect to the important role of the large cations: Pb2+, Bi3+or Ba2+. A number of iodates with similar structures are also analyzed.

some additional typos :

p1 line39 on  a par -> in line
p2 line3 realize -> unravel
p2 line66 out at temperature 543 K -> out at 543 K

p5 line178 dessapeared -> disappeared
p5 line183 cofirmed -> confirmed
p5 line184 twinnning -> twinning
p5 line186 orthorombic -> orthorhombic
p5 line189 accrodingly -> accordingly

p14 line350 twinnig -> twinning

Author Response

We are very thankful to the Reviewer for accurate reading of our manuscript and valuable suggestions.

We agree with the Reviewer that our angles are given with small deviation of 90° for α and β-angles as they were in our experiment at low temperature. We made before and cheсked now our refinements with the angles equal to 90°. It doesn't change the result. It is connected with the usually overestimated values of deviation. The same often concern atomic coordinates   which because of physical reasons can’t possess such small value as sometime 0.00001. We send our .cif in CCDC  as it is in manuscript thus we did not change α and β-angles. We tried to solved structure in triclinic unit cell, however the model was not completed. Only using of monoclinic approximation allowed finding the model. It was not discussed in manuscript.

The Abstract content looks much better thanks Reviewer’s  suggestion and we accepted all of them.  We are also very thankful for corrections of our sudden typos which we missed. Also we tried as much as possible to improve our English with the help of person known it much better than we. We corrected also some other typos.
